# Pain Prevalence and Satisfaction with Pain Management in Inpatients: A Cross-Sectional Study

**DOI:** 10.3390/healthcare11243191

**Published:** 2023-12-18

**Authors:** Ángel Becerra-Bolaños, Annette Armas-Domínguez, Lucía Valencia, Pedro Jiménez-Marrero, Sergio López-Ruiz, Aurelio Rodríguez-Pérez

**Affiliations:** 1Department of Anesthesiology, Intensive Care and Pain Medicine, Hospital Universitario de Gran Canaria Doctor Negrín, 35010 Las Palmas de Gran Canaria, Spainarodperp@gobiernodecanarias.org (A.R.-P.); 2Department of Medical and Surgical Sciences, Universidad de Las Palmas de Gran Canaria, 35001 Las Palmas de Gran Canaria, Spain

**Keywords:** pain, quality of healthcare, patient satisfaction, acute pain service, pain management, pain monitoring

## Abstract

Background: Pain in hospitalized adults is underestimated and undervalued. The aim of this study was to evaluate pain prevalence and satisfaction with the hospital’s pain management among patients attending a tertiary university hospital. Predictor factors of pain were also studied. Methods: A prospective, cross-sectional study was carried out through a structured questionnaire given on one day to all hospitalized patients in a university hospital. Clinical data, such as personal history and analgesic treatment, were collected from medical records. Other variables related to pain (including intensity rated by the visual analogue scale as well as location and patient satisfaction measured by the numerical rating scale) were also obtained. Results: Of the 274 surveyed patients, pain prevalence was 52.9%, with an average intensity of 5.3 ± 2.8 according to VAS. The overall satisfaction was 87.2%, and 72.6% had already been prescribed at least one analgesic. Patients receiving analgesics showed higher pain intensity (VAS 3.6 ± 3.4) than those without treatment (VAS 1.1 ± 2.1) (*p* < 0.001). However, patients with treatment showed more satisfaction (NRS 7.8 ± 2 vs. 5.3 ± 1.4, *p* < 0.001). Conclusions: The prevalence of pain in hospitalized patients was high, despite the fact that patient satisfaction was also very high.

## 1. Introduction

Pain constitutes a frequent concern in a hospital environment. Nevertheless, in daily clinical practice, not enough attention is given to the presence of pain in hospitalized patients, and many patients report suffering pain during their hospital stay [1]. This may be due to the high care burden, either because pain is not the reason for consultation or because it is a concurrent symptom manifesting with those caused as a consequence of the presenting disease. Unfortunately, the treatment of pain is sometimes passed over in favor of diagnosing the primary disorder. Moreover, patients often avoid complaining about suffering from pain, perceiving it as something inevitably associated with hospital medical care [2]. In addition, during the COVID-19 pandemic, the fear of contagion and the work overload may have limited doctor–patient interviews [3] and consequently may have reduced the time spent on the exploration of pain. All these reasons may have increased the prevalence of pain as a consequence of its underestimation and undertreatment.

Pain in hospitalized patients is, and should be, one of the main concerns from a therapeutic point of view. Inadequate pain management is associated with significant costs and socioeconomic problems. This is due to the fact that it causes a delay in recovery, a longer hospital stay, and an increase in healthcare costs [4]. Inadequate pain management also leads to negative consequences that affect personal, family, economic, and social well-being. This, in turn, provokes a detrimental effect on patients’ quality of life [5,6]. Thus, access to pain treatment is considered a fundamental human right.

Our main objective was to evaluate the prevalence of pain in patients admitted to a tertiary university hospital. We also evaluated the satisfaction of patients regarding pain management during their hospital stay. As a secondary objective, we evaluated the percentage of patients with pain who were receiving analgesics. We also analyzed different risk factors that may have influenced pain during the hospital stay.

## 2. Materials and Methods

This prospective, cross-sectional study evaluated the prevalence of pain using a questionnaire given on one day to all hospitalized patients in a tertiary university hospital. The study protocol and questionnaire were approved by the Ethics Committee of the Hospital Universitario de Gran Canaria Doctor Negrín, Las Palmas de Gran Canaria, Spain (IRB approval #2021-184-1), and prospectively registered at ClinicalTrials.gov (NCT04901650). The research was carried out according to the Declaration of Helsinki. The manuscript follows the STROBE guidelines [7]. The STROBE initiative establishes recommendations regarding the items that should be addressed in articles reporting cross-sectional studies. Written, informed consent from each patient was obtained before enrollment. All adult patients admitted to the hospital for more than 24 h were included in the study. Exclusion criteria were: patients with cognitive impairment, difficulty with comprehension, speech or language barriers, psychiatric disorders, patients admitted to the intensive care unit (ICU), those with COVID-19, and patients who refused to participate.

On 8 June 2021, eighteen members of the anesthesiology department from the Hospital Universitario de Gran Canaria, Doctor Negrín, visited all of the wards to carry out the survey. The hospital in which the study was carried out is a tertiary university hospital located in a provincial capital. This serves as a reference for the following medical specialties: internal medicine, cardiology, pulmonology, endocrinology, nephrology, rheumatology, digestive, neurology, medical oncology, radiation oncology, geriatrics, hematology, and psychiatry. It also has the following surgical specialties: vascular surgery, cardiovascular surgery, otorhinolaryngology, plastic surgery, general and digestive surgery, thoracic surgery, neurosurgery, ophthalmology, traumatology and orthopedic surgery, maxillofacial surgery, and urology. All named specialties are capable of training new specialists through an accredited system of specialized health training. As it is a hospital that is part of the Spanish public health system, the sociocultural and educational characteristics of the population served are heterogeneous. 

It was decided to restrict data collection to a homogeneous population, in which there were no communication problems. For this reason, patients from the ICU and those with communication and cognitive difficulties were excluded. Due to the pandemic situation and the methodology of the study (data collection through personal surveys), we considered that excluding COVID-19 patients was the most prudent course in order to carry out the study, thereby avoiding the spread of the virus among the rest of the hospital patients and the healthcare workers. Although psychiatric patients may have been especially vulnerable to pain, the interview in this population can be more complicated due to fluctuations in psychiatric pathology. No deviation from the original protocol occurred during the course of the study. 

The survey consisted of three sets of questions (Appendix A). The first set collected information on the characteristics of each patient, such as age, gender, comorbidities, occupation, marital status, and other personal antecedents that could be related to the presence of pain. The second set dealt with the presence of pain, where it was located, and its intensity. The Visual Analogue Scale (VAS) was employed to evaluate pain intensity, from 0 (no pain) to 10 (maximum pain), based on the patient indicating a point on a horizontal line of 10 cm in length representing his experience of pain. This allowed for a measurement of the intensity with maximum ability for reproduction among the observers. This intensity was expressed in centimeters. The pain intensity was also stratified as follows: mild pain if the patient indicated VAS < 3, moderate pain if VAS was between 4 and 7, and severe pain when VAS ≥ 8. The third set explored the level of satisfaction in terms of pain management during their hospital stay. Satisfaction was measured not only as a dichotomous variable (yes/no), but also according to a Numerical Rating Scale (NRS) from 0 (no satisfaction) to 10 (maximum satisfaction). Therefore, both data on pain and patient satisfaction with the hospital’s pain management can be analyzed from a qualitative point of view: patient with or without pain, and patient satisfied or not. By using these scales, data can also be analyzed from a quantitative point of view: in terms of pain, using the VAS scale from 0 to 10, and in terms of satisfaction, using the NRS scale from 0 to 10.

The fact that surveys were conducted by medical personnel experienced in pain management and accustomed to clinical interviewing ensured that the survey would be conducted under the best clinically available conditions.

Analgesic treatment protocols administered during the hospital stay, as well as clinical data (comorbidities and demographic data), were collected after reviewing electronic medical records. 

### Statistical Analysis

Data on categorical variables were expressed as frequency and percentage. Quantitative variables were expressed as the mean ± SD. A chi-square test was used to compare frequency data between groups. We used the Shapiro–Wilk test to analyze the normality of the data. To compare quantitative variables between two groups, a *t*-test for independent samples was used in cases of variables with a normal distribution and a Mann–Whitney U test when the distribution of variables could not be adjusted to normality. To compare quantitative variables among more than two groups, a one-way analysis of variance (ANOVA) test for independent samples was used for variables with a normal distribution and a Kruskal–Wallis test when the distribution was not adjusted to normality. A *p* value < 0.05 was considered statistically significant. The data were analyzed using SPSS 24.0 (Statistical Package for Social Sciences, IBM, Chicago, IL, USA).

## 3. Results

Of the 544 patients admitted for more than 24 h on the abovementioned day, the survey was conducted on 274. The exclusion criteria for the other 270 were: 41 admitted to the ICU, 8 infected with COVID-19, 32 admitted to the psychiatry ward, 109 had some form of communication problem, and 80 rejected the survey (Figure 1). Characteristics of the patients surveyed are shown in Table 1.

The prevalence of pain detected from the survey was 52.9% (145 patients). Those patients who suffered pain reported an average intensity of 5.3 ± 2.8, according to VAS. The most frequent pain occurred in the lower limbs (50 patients, 34.5%), abdomen (26 patients, 17.9%), lumbar/spine (23 patients, 15.9%), thorax (19 patients, 13.1%), head (12 patients, 8.3%), pelvic region (8 patients, 5.5%), and upper limbs (7 patients, 4.8%). The pain prevalence in the non-surgical wards of the hospital was 50.3% (91/181 patients) with an intensity of 3.0 ± 3.5. On the surgical wards, the prevalence was somewhat higher (58%, 54/93 patients), although the intensity was slightly lower (2.7 ± 2.8). Table 2 and Table 3 compare the characteristics of patients and admission among those suffering pain as opposed to those not suffering pain. Figure 2 shows the prevalence of pain according to the service in charge of patient care.

Of the 274 surveyed patients, 87.2% were satisfied with the pain management received, with an average rating of 7.1 ± 2.2 according to the NRS. Of those patients who experienced pain, 77.2% (112/145) reported feeling satisfied with the pain management during their hospital stay. This percentage was lower than that indicated by those patients who did not suffer pain (98.4%, 127/129); this difference was statistically significant (*p* < 0.0001). However, when assessing satisfaction as a quantitative variable, according to the NRS reported by patients, we did not find differences. Patients suffering from pain reported a mean satisfaction of 6.9 ± 2.3 according to the NRS, while those who did not report pain showed a mean satisfaction of 7.4 ± 2.1 (*p* = 0.070). When analyzing satisfaction according to gender, we detected that 85.8% of men and 88.9% of women were satisfied with analgesic management (*p* = 0.447). Data between surgical and non-surgical patients and their satisfaction with pain management and the application of analgesia during hospital admission are shown in Figure 2.

Of the patients surveyed, 72.6% (199 patients) had already been prescribed at least one analgesic. This percentage was slightly higher in patients who suffered pain (84.8%) than in those without pain (58.9%). Moreover, patients who were receiving analgesics showed a higher pain intensity than those who were not (Table 4). On the other hand, we did not detect differences in the percentage of satisfaction reported by patients according to whether treatment was administered or not. However, those patients who were receiving analgesics showed a higher degree of satisfaction (Table 5). We detected that the administration of the different analgesics and coadjuvant analgesic drugs (antidepressants and neuroleptics) was higher in patients with pain. However, we observed no significant difference in the presence of pain in terms of anxiolytic administration. Consistently, we found no differences between the administration of the different drugs and the degree of satisfaction (Table 6). Upon analyzing the administration of analgesics and the characteristics of admission, 74.6% (156/209) of patients admitted to the emergency service received analgesic treatment as opposed to 66.2% (43/65) of those admitted electively (*p* = 0.180). We also found no differences between the percentage of patients who were receiving treatment and the length of hospital stay: less than three days (68.6%), 4–15 days (75.2%), or more than 14 days (70.7%) (*p* = 0.601). However, we did find statistically significant differences regarding the administration of treatment and the service in charge of hospital admission: 90.3% (84/93) of the patients on surgical wards received analgesics, as opposed to 63.5% (115/181) (*p* < 0.0001) of those on non-surgical wards. The percentage of men receiving treatment was similar to that detected in women (68.9% vs. 76.9%, respectively; *p* = 0.136). Moreover, the evolution of pain during admission was similar with respect to gender: 57.4% of men reported that pain had decreased, compared to 64.3% of women (*p* = 0.221).

## 4. Discussion

In this cross-sectional observational trial carried out in a tertiary university hospital, more than half of the admitted patients suffered pain, regardless of patient characteristics or conditions of their admission. This prevalence is similar to that detected by some studies, which show a prevalence of pain of 48.5–63% [8,9,10,11,12]. However, other studies show pain prevalence above 69.5% [13,14], and as much as above 90% [15]. The differences among published studies may be due to the different tools used to detect the presence of pain: some studies use the NRS [15], others collect it qualitatively [13], and others use specific instruments such as the Brief Pain Inventory [14]. It is also necessary to take into account that collecting data through surveys can be influenced by the cultural characteristics of patients. In any case, the purpose of these studies is not to compare the prevalence of pain among hospitals or countries but to find out the prevalence in a specific population in order to reduce it. From all these studies, it can be concluded that the prevalence of pain in hospitals is higher than acceptable [14,16,17]. All patients under medical/healthcare supervision should have their pain assessed and managed following evidence-based practice. Thus, both the evaluation and treatment of pain must be carried out on an individualized basis with the aim of minimizing the patient’s painful experience as much as possible.

It is necessary to establish a plan to improve analgesic management in our patients and to perform an adequate screening, monitoring pain as the fifth vital sign [18]. A personal evaluation of pain is necessary to provide adequate treatment, which will have a positive impact on the duration and quality of the hospital stay [19]. The use of simple scales, such as the Visual Analogue Scale, has gained popularity, allowing for the detection of pain [20]. Some authors believe that monitoring is complicated because it is not measured objectively and is based solely on a subjective sensation [21]. However, it is necessary to carry out an adequate diagnosis and a prior evaluation in order to establish methods to provide relief.

Of the participants who were receiving analgesic treatment, 61.8% reported having more intense pain. Analgesics do not always relieve pain completely. This could be due to an overestimation of the therapeutic efficacy of analgesic medicines on the part of the physician [16]. In a study carried out by Strohbuecker et al., they confirmed that patients with more intense pain receive more analgesics. This can indicate that there is infra-therapeutic management for pain [12]. However, given the distribution of analgesics between those with pain and those without pain in our study, it can be inferred that treatment had been prescribed to those patients who had reported pain.

Although more than half of the surveyed patients presented with pain, 87.2% were satisfied with the analgesic management. This has already been published [10,22]. Previous publications point out that the high level of satisfaction is independent of the analgesics prescribed [23,24]. In our study, we found no differences in the percentage of satisfied patients according to the analgesic regimen. The high level of satisfaction in spite of the high prevalence of pain could be considered contradictory. On the one hand, satisfaction may be affected by patient expectations, knowledge about the characteristics of hospital admission, and beliefs about pain or resignation [10,22,23]. On the other hand, patients’ responses to the questions asked by healthcare providers (even though they were not the prescribing physician) may be influenced by factors unrelated to the aim of the study, such as fear of negative repercussions in the treatment they are going to receive or fear of causing any damage to the reputation of the doctor and/or the health system. Nevertheless, given the subjective nature of pain, the patient’s own perception of pain management throughout the hospital admission is of great importance.

There is some controversy regarding the presence of risk factors that predispose a patient to a more severe intensity of pain. Some studies have indicated that females are more predisposed to a higher prevalence of pain [5,8,11,25,26,27]. There could be a biological or psychosocial basis that explains the differences regarding gender, the perception of pain, or the response to treatment [28]. In our study, we did not find differences in the prevalence of pain in reference to gender, as has been previously published [12,13,23,29]. Nor did we detect a relationship between pain and the age of patients [10,12,15,23]. Other authors have found that older patients present with less pain [8,13,26], possibly due to the fact that this population has a higher tolerance to pain and resigns themselves to it, assuming pain to be an inevitable part of growing old [30]. Also, the degree of demyelination of pain pathways could contribute to this effect. The comorbidities included in the study were not found to affect the prevalence or intensity of pain. However, patients suffering from more than one comorbidity reported suffering more pain. Also, there is a significant positive correlation between the number of comorbidities and the intensity of pain. Unlike previous studies [31,32], we did not find any relationship between anxiety and pain, probably because the survey included anxiety only as a personal antecedent and no specific anxiety scale was used. In regard to other patients’ characteristics that may predispose to greater pain suffering, we did not find differences regarding occupation, marital status [23,30], educational level [8,10,11,23,30], or religion. Although there is no previous publication that correlates them, there could be a relationship between religion and pain. Religious or spiritual practices can help manage pain, but suffering from pain can have a different meaning for those patients with fervent religious beliefs, increasing their tolerance [33].

We acknowledge some potential limitations of this study. First, the analysis is based on a survey carried out by doctors. The presence of a doctor during the survey, although not the treatment doctor, could have influenced the patients’ report about their level of satisfaction. Although patients with cognitive or comprehension disorders were excluded, some participants may have been confused by one or more of the questions. Second, the questionnaire used is self-made and not validated. However, the pragmatic perspective of the study made it necessary for the questionnaire to be based on usual clinical practice. Anesthesiologists performing the survey have extensive experience in pain management and are accustomed to clinical interviewing, ensuring the best clinically available conditions. Therefore, the questionnaire used was simple enough to be generalizable and understandable by all patients. Questions included in the survey have content validity by nature. Specifically, the questions about pain intensity and satisfaction with the treatment received are entirely subjective, and there is little possibility that they can be answered in any other way. In addition, pain in this study was assessed as a dichotomous qualitative variable (yes or no) and as a quantitative variable (VAS). While pain treatment is not one-size-fits-all, the need to use a method to measure pain intensity should be mandatory for everyone. In addition, clinicians should adapt the pain assessment, using the most appropriate tool for the patient’s clinical situation. Third, our study was carried out at only one center, so the number of patients was limited. We excluded critically ill patients and patients with potential communication disorders. In addition, our center does not have pediatric and obstetric patients, so the data obtained cannot be extrapolated to these types of patients. In addition, to include a larger and more diverse population and reach more convincing conclusions, it would have been interesting to conduct the survey on different days over several months. In future studies, we will increase the survey days to reach a larger population and reach more convincing conclusions.

## 5. Conclusions

Pain in hospital patients is an important issue, even if patients are satisfied. Pain cannot be treated if it has not been previously evaluated. Therefore, it is important to adequately and repeatedly check the patient’s pain during the hospital stay to evaluate the evolution of pain intensity and the efficacy of its treatment. The continuous recording of this variable increases the awareness of the patients’ experience of pain.

## Figures and Tables

**Figure 1 healthcare-11-03191-f001:**
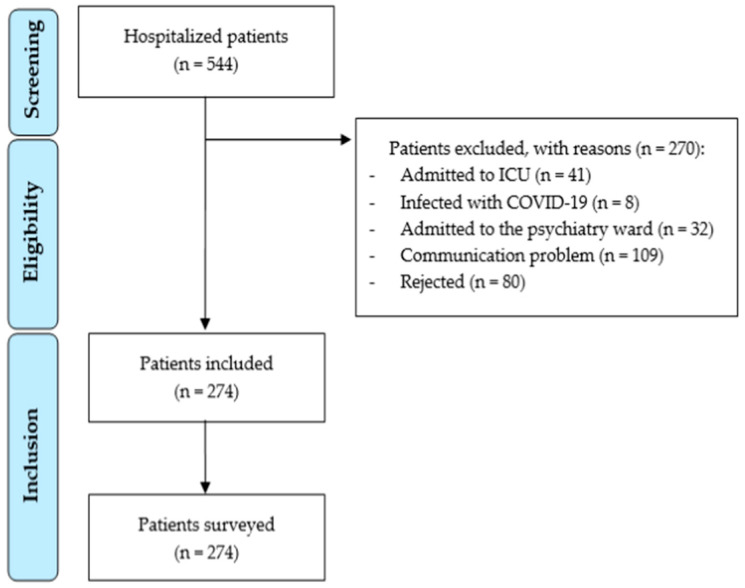
Patient flow chart diagram.

**Figure 2 healthcare-11-03191-f002:**
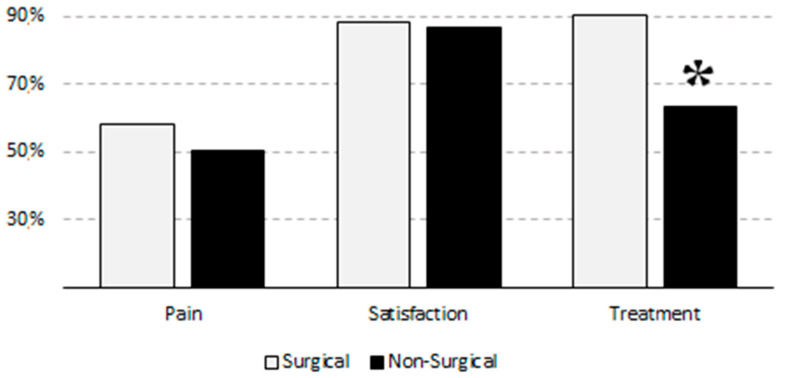
Pain, satisfaction regarding pain management, and application of analgesia during hospital admission according to the service in charge of patient care. Data are expressed as percentages. Treatment = analgesics; * *p* < 0.0001.

**Table 1 healthcare-11-03191-t001:** Characteristics of patients and hospital admissions.

		*n* = 274
Gender female, *n* (%)		126 (46)
Age, years		66.17 ± 14.8
Race, *n* (%)	Caucasian	272 (99.3)
	Non-Caucasian	2 (0.7)
Marital status, *n* (%)	Single	41 (14.9)
	In a relationship/Married	177 (64.6)
	Divorced/Widowed	56 (20.4)
Religion, *n* (%)	Atheist	46 (16.8)
	Christian	218 (79.6)
	Muslim	1 (0.4)
	Others	9 (3.3)
Employment situation, *n* (%)	Employed	124 (45.2)
	Retired	118 (43.1)
	Housework	21 (7.7)
	Unemployed	11 (4.0)
Comorbidities, *n* (%)	Arterial hypertension	133 (48.5)
	Diabetes Mellitus	104 (37.9)
	Dyslipidemia	84 (30.6)
	Rheumatic disease	24 (8.7)
	Respiratory disease	47 (17.1)
	Ischemic cardiopathy	33 (12.0)
	Chronic renal failure	19 (6.9)
	Anxiety	24 (8.8)
Service in charge of patient care, *n* (%)	Non-surgical	181 (66.1)
	Surgical	93 (33.9)
Length of hospital stay, *n* (%)	Less than 3 days	51 (18.6)
	4–14 days	141 (51.5)
	More than 14 days	82 (29.9)
Admission through the emergency department, *n* (%)		209 (76.3)

Data are expressed as frequency (percentage) or mean ± SD.

**Table 2 healthcare-11-03191-t002:** Prevalence and intensity of pain according to the characteristics of patients.

		Pain		*p*	Pain Intensity	*p*
		No (*n* = 129)	Yes (*n* = 145)			
Gender, *n* (%)	Female	53 (41.1)	73 (50.3)	0.125	2.6 ± 3.1	0.094
	Male	76 (58.9)	72 (49.7)		3.3 ± 3.5	
Age, years		67.9 ± 13.4	64.7 ± 15.8	0.073	Corr −0.074	0.223
Race, *n* (%)	Caucasian	129 (100)	143 (98.6)	0.408	2.9 ± 3.3	0.788
	Non-Caucasian	0 (0)	2 (1.4)		3.5 ± 1.5	
Marital status, *n* (%)	In a relationship/married	83 (64.3)	94 (64.8)	0.418	2.9 ± 3.3	0.815
	Single/divorced/widowed	46 (35.7)	51 (35.2)		3.1 ± 3.3	
Religious, *n* (%)	Yes	108 (83.7)	120 (82.7)	0.721	3.0 ± 3.4	0.793
	No	21 (16.3)	25 (17.2)		2.7 ± 2.9	
Comorbidities, *n* (%)	Arterial hypertension	59 (45.7)	74 (51)	0.381	3.3 ± 3.1	0.065
	Diabetes Mellitus	41 (31.8)	63 (43.4)	0.102	3.4 ± 3.4	0.051
	Dyslipidemia	35 (27.1)	49 (33.8)	0.233	3.2 ± 3.4	0.254
	Rheumatic disease	10 (7.7)	14 (9.7)	0.578	3.1 ± 3.2	0.786
	Respiratory disease	19 (14.7)	28 (19.3)	0.315	3.3 ± 3.3	0.434
	Ischemic cardiopathy	14 (10.9)	19 (13.1)	0.568	3.5 ± 3.6	0.329
	Chronic renal failure	12 (9.3)	17 (11.7)	0.515	3.1 ± 3.3	0.782
	Anxiety	10 (7.7)	14 (9.7)	0.578	3.0 ± 3.3	0.781
Number of comorbidities, units		1.52 ± 1.28	1.90 ± 1.52	0.025	Corr 0.138	0.023

Data are expressed as frequency (percentage) or mean ± SD. Corr: Pearson correlation.

**Table 3 healthcare-11-03191-t003:** Prevalence and intensity of pain according to the characteristics of hospital admission.

		Pain		*p*	Pain Intensity	*p*
		No (*n* = 129)	Yes (*n* = 145)			
Admission, *n* (%)	Emergency	95 (73.6)	114 (78.6)	0.334	3.1 ± 3.4	0.057
	Scheduled	34 (26.4)	31 (21.4)		2.2 ± 2.8	
Service in charge of patient care, *n* (%)	Non-surgical	90 (69.8)	91 (62.8)	0.221	3.0 ± 3.5	0.498
	Surgical	39 (30.2)	54 (37.2)		2.7 ± 2.8	
Length of hospital stay, *n* (%)	Less than 3 days	23 (17.8)	28 (19.3)	0.952	2.8 ± 3.3	0.783
	4–14 days	67 (51.9)	74 (51.0)		2.8 ± 3.2	
	More than 14 days	39 (30.2)	43 (29.6)		3.1 ± 3.5	

Data are expressed as frequency (percentage) or mean ± SD.

**Table 4 healthcare-11-03191-t004:** Prevalence and intensity of pain according to the treatment prescription.

		Pain		*p*	Pain Intensity	*p*
		No (*n* = 129)	Yes (*n* = 145)			
Treatment	Yes (*n* = 199)	76	123	<0.001	3.6 ± 3.4	<0.001
	No (*n* = 75)	53	22		1.1 ± 2.1	

Data are expressed as frequency or mean ± SD.

**Table 5 healthcare-11-03191-t005:** Satisfaction according to the treatment prescription.

		Satisfaction		*p*	Satisfaction Degree	*p*
		No (*n* = 35)	Yes (*n* = 239)			
Treatment	Yes (*n* = 199)	29	170	0.146	7.8 ± 2.0	<0.001
	No (*n* = 75)	6	69		5.3 ± 1.4	

Data are expressed as frequency or mean ± SD.

**Table 6 healthcare-11-03191-t006:** Pain and satisfaction according to the different drugs prescribed.

	Pain		*p*	Satisfaction		*p*
	No (*n* = 129)	Yes (*n* = 145)		No (*n* = 35)	Yes (*n* = 239)	
Weak opioids, *n* (*%*)	14 (10.8)	34 (23.4)	0.006	7 (20.0)	41 (17.2)	0.679
Strong opioids, *n* (*%*)	5 (3.9)	26 (17.9)	<0.001	7 (20.0)	24 (10.0)	0.082
Opioids + NSAIDS, *n* (*%*)	17 (13.2)	43 (29.6)	0.001	11 (31.4)	49 (20.5)	0.144
Anxiolytics, *n* (*%*)	21 (16.3)	31 (21.4)	0.283	4 (11.4)	48 (20.1)	0.223
Antidepressants, *n* (*%*)	0 (0)	7 (4.8)	0.011	0 (0)	7 (2.9)	0.305
Neuroleptics, *n* (*%*)	6 (4.6)	15 (10.3)	0.077	3 (8.6)	18 (7.5)	0.829

Data are expressed as frequency and percentage.

## Data Availability

The data presented in this study are available on request from the corresponding author.

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
