# Peer review of "Pain Prevalence and Satisfaction with Pain Management in Inpatients: A Cross-Sectional Study"

_healthcare, 2023, doi:10.3390/healthcare11243191_

Round 1
Reviewer 1 Report
Comments and Suggestions for Authors
Thank you for the opportunity to review this paper investigating the prevalence of pain in in-patients at a tertiary hospital.
The study offers some insight into the current prevalence of pain in a census-type study, along with patient satisfaction with their care. Interpreting this data is made difficult by insufficient details about the nature of the hospital setting. For example, while the wards are described as ‘surgical’ or ‘non-surgical’ and it is stated that obstetric and paediatric wards are not part of this hospital, there was no description of the range of clinical specialties nor the reason for admission. Patients in ICU, those with communication and cognitive difficulties, those with Covid-19, and psychiatric patients were not included, along with those who did not wish to participate. The omission of these patients (excluding those not wanting to take part) is disappointing as those admitted with psychiatric problems in particular may well have reported pain, and that their pain was not managed.
The authors report comorbidities but did not describe multi-morbidities that could elevate the risk of pain. An analysis of the number of comorbidities each patient presented with would be helpful.
The authors also indicate that they did not find differences in pain prevalence relating to gender, but did not provide breakdowns related to gender for provision of analgesia, nor response to analgesia or satisfaction. Undertaking these analyses is critical because of the known disparity in analgesia use when women present with pain as compared with men.
The discussion was a little confusing for me. The authors identify that various studies have shown different pain prevalence amongst their populations and suggest that this could be due to the measurement tools employed or the cultural characteristics of those included. They do not mention the type of hospital, the clinical specialties in that hospital, the socio-political and health policy factors that might influence hospital admission, yet these are well-established contributors to treatment seeking and hospital admission.
The authors also point out that their study relied on verbal responses from patients, and go into some detail about the various measures that could be employed in those who are either critically ill, sedated or have communication difficulties. These instruments were not used, and I find it difficult to understand the relevance of this section in the discussion. Given that pain is a subjective sensory and emotional experience, measures of pain behaviour are just that: measures of pain behaviour, and not necessarily well-correlated with verbal report, or with actual experience.
While the authors do acknowledge their survey has not been validated, and that medical practitioners undertook the survey, introducing potential demand characteristics on patient responses. They acknowledge that some patients may have found the questions difficult to understand, but did not take steps to pilot the survey before the study. Additionally, some of the factors associated with greater pain reporting, such as anxiety, were not measured using validated instruments nor collected at the time patients were asked to report their pain intensity. There is no copy of the survey, and the authors do not clearly identify the ‘personal antecedents’ that they suggest could influence pain reporting.
Author Response
Ref: Submission ID healthcare-2748282: “Pain prevalence and satisfaction with pain management in inpatients: a cross-sectional study”
Reviewer #1
Q1) Thank you for the opportunity to review this paper investigating the prevalence of pain in in-patients at a tertiary hospital.
R1.- Thank you very much for the comments regarding the manuscript. We believe that suggested comments have been key to improving the manuscript.
Q2) The study offers some insight into the current prevalence of pain in a census-type study, along with patient satisfaction with their care. Interpreting this data is made difficult by insufficient details about the nature of the hospital setting. For example, while the wards are described as ‘surgical’ or ‘non-surgical’ and it is stated that obstetric and paediatric wards are not part of this hospital, there was no description of the range of clinical specialties nor the reason for admission. Patients in ICU, those with communication and cognitive difficulties, those with Covid-19, and psychiatric patients were not included, along with those who did not wish to participate. The omission of these patients (excluding those not wanting to take part) is disappointing as those admitted with psychiatric problems in particular may well have reported pain, and that their pain was not managed.
R2.- Thank you very much for the opportunity to clarify this issue.
We acknowledge that it would have been interesting to show data from ICU patients. However, we decided to restrict data collection to a homogeneous population, in which there were no communication problems. For this reason, patients with communication and cognitive difficulties were also excluded. Furthermore, due to the pandemic situation and the methodology of the study (data collection through personal surveys), we considered that excluding Covid-19 patients was the most prudent course of action in order to carry out the study and avoid the spread of the virus among the rest of the hospital patients and the healthcare workers. Regarding psychiatric patients, we completely agree with the reviewer that this population may have been especially vulnerable to pain. However, the interview in this type of patient can be more complicated. Furthermore, due to these fluctuations in psychiatric pathology, we considered that this population should be excluded to optimize data collection in the rest of the hospitalization wards.
We have added this information to the Material and Methods section (lines 84 – 92).
Q3) The authors report comorbidities but did not describe multi-morbidities that could elevate the risk of pain. An analysis of the number of comorbidities each patient presented with would be helpful.
R3.- We truly appreciate this comment. Following the Reviewer’s suggestion, we have performed the appropriate statistical analysis and added this information in Table 2.
Q4) The authors also indicate that they did not find differences in pain prevalence relating to gender, but did not provide breakdowns related to gender for provision of analgesia, nor response to analgesia or satisfaction. Undertaking these analyses is critical because of the known disparity in analgesia use when women present with pain as compared with men.
R4.- Thank you very much for this suggestion. We have performed the analysis and added this information to Results section:
“When analyzing satisfaction according to gender, we detected that 85.8% of men and 88.9% of women were satisfied with analgesic management (p = 0.447).” (Lines: 164 – 166)
“The percentage of men receiving treatment was similar to that detected in women (68.9% vs 76.9%, respectively; p = 0.136). Moreover, the evolution of pain during admission was similar with respect to gender: 57.4% of men reported that pain had decreased, compared to 64.3% of women (p = 0.221).” (Lines: 184 – 187)
Q5) The discussion was a little confusing for me. The authors identify that various studies have shown different pain prevalence amongst their populations and suggest that this could be due to the measurement tools employed or the cultural characteristics of those included. They do not mention the type of hospital, the clinical specialties in that hospital, the socio-political and health policy factors that might influence hospital admission, yet these are well-established contributors to treatment seeking and hospital admission.
R5.- We sincerely thank the reviewer for this suggestion. As requested, we have included the following information in the Materials and Methods section, to give context to the information obtained in the study:
“The hospital in which the study was carried out is a tertiary university hospital located in a provincial capital. This serves as a reference for the following medical specialties: internal medicine, cardiology, pulmonology, endocrinology, nephrology, rheumatology, digestive, neurology, medical oncology, radiation oncology, geriatrics, hematology and psychiatry. It also has the following surgical specialties: vascular surgery, cardiovascular surgery, otorhinolaryngology, plastic surgery, general and digestive surgery, thoracic surgery, neurosurgery, ophthalmology, traumatology and orthopedic surgery, maxillofacial surgery, and urology. All named specialties are capable of training new specialists through an accredited system of specialized health training. As it is a hospital that is part of the Spanish public health system, the sociocultural and educational characteristics of the population served are heterogeneous.” (Lines: 73 – 83)
Q6) The authors also point out that their study relied on verbal responses from patients, and go into some detail about the various measures that could be employed in those who are either critically ill, sedated or have communication difficulties. These instruments were not used, and I find it difficult to understand the relevance of this section in the discussion. Given that pain is a subjective sensory and emotional experience, measures of pain behaviour are just that: measures of pain behaviour, and not necessarily well-correlated with verbal report, or with actual experience.
R6.- We agree with what is stated by the Reviewer, given that pain is a subjective sensory and an emotional experience. As suggested, we have decided to delete the paragraph mentioned.
Q7) While the authors do acknowledge their survey has not been validated, and that medical practitioners undertook the survey, introducing potential demand characteristics on patient responses. They acknowledge that some patients may have found the questions difficult to understand, but did not take steps to pilot the survey before the study. Additionally, some of the factors associated with greater pain reporting, such as anxiety, were not measured using validated instruments nor collected at the time patients were asked to report their pain intensity. There is no copy of the survey, and the authors do not clearly identify the ‘personal antecedents’ that they suggest could influence pain reporting.
R7.- We appreciate the observations made by the reviewer.
We have added the following explanation in the Material and Methods section: “The fact that surveys were conducted by medical personnel experienced in pain management and accustomed to clinical interviewing ensured that the survey would be conducted under the best clinically available conditions." (Lines: 109 – 111)
The personal antecedents studied as potential risk factors are shown in Table 2 (comorbidities, religiosity, race or marital status). However, it was not detected that comorbidities influenced the suffering of pain or its greater intensity. As suggested by the Reviewer, we have added the following to the Discussion section: "The comorbidities included in the study were not found to affect the prevalence or intensity of pain. However, patients suffering from more than one comorbidity reported suffering more pain. Also, there is a significant positive correlation between number of comorbidities and intensity of pain." (Lines: 274 – 277)
Reviewer 2 Report
Comments and Suggestions for Authors
Peer review report for the manuscript entitled “Pain prevalence and satisfaction with pain management in in-2 patients: a cross-sectional study.”
The study determined the pain prevalence in a tertiary university hospital in Spain, the risk factors of the pain, and patient satisfaction with the hospital’s pain management. The results showed that pain prevalence among adult in-patients was high, no risk factor examined was significantly associated with pain, and patients were satisfied with the hospital’s pain management despite the high pain intensity.
I have suggestions and comments to improve the manuscript and hope the authors will consider them.
1. Line 14. I suggest the authors clarify the “satisfaction” with the hospital’s pain management.
2. Line 25. “A problem yet to be solved” gives me the impression that the hospital has tried to solve the issue but has yet to succeed. However, the current study actually just explored the pain prevalence. Thus, I suggest rephrasing to accurately conclude from the study, which showed that the pain prevalence is high. (While hospitals worldwide strive to provide “pain-free” environments, 52.9% is highly prevalent.)
3. Lines 53 – 55. I do not understand the sentence, so I suggest rephrasing it.
4. The Introduction Section needs improvement. Pain management has been studied around the globe for the past decades. Why is it crucial for the authors to conduct the study?
5. Line 63. Perhaps the authors can briefly describe (1 or 2 sentences) STROBE guidelines, such as what STROBE stands for and its uses and aims.
6. Lines 68 – 69. The sentence is a duplicate of what is in lines 63 – 64.
7. Lines 77 – 80. I suggest rephrasing for clarity. (Is the horizontal line 10 cm in length?)
8. Lines 85 – 86. Perhaps the authors should provide the reason for NRS to be included. (qualitative & quantitative data?)
9. Line 102. I suggest rephrasing for clarity. (Have 544 patients been hospitalized for more than 24 hours or recently admitted within 24 hours?
10. The tables may need to be rearranged for better visualization.
11. Line 136. “Thus” does not seem to be the right word.
12. Line 166. The in-text citation.
13. Lines 171 – 173. The authors may need to follow up with the sentence. What is the purpose of this study? what is the population?
14. Line 174. What is the acceptable prevalence of pain, then?
15. Line 175 – 177. This statement seems to belong in the introduction section. Why it is what it is to convince the importance of this study.
16. Lines 185 – 210. I do not see this paragraph as relevant. The patients who are best candidates for other methods are excluded from this study.
17. Line 211. I suggest replacing “complained” with “reported.”
18. Lines 216 – 218. Perhaps this paragraph presented one of the study's limitations. while pain treatment is not one-size-fits-all, the method to measure pain intensity is.
19. Lines 225 – 226. I don't think the statement is valid unless the questions survey the pain improvement with treatment rather than the pain present.
20. Line 243. Suggest replacing “determined” with “found.”
21. Lines 256 – 261. Suggest rearranging the sentence. The sentence from lines 257 through 259 may be better following the sentence from lines 260 through 261—the second limitation.
22. Lines 260 – 269. The justification for the second limitation is lengthy and unreasonable. Without validation, perhaps, the questionnaire should be piloted before being distributed.
23. Line 275. “Unsolved problem”. I refer to my comment on this phrase within the abstract.
Comments on the Quality of English LanguageEven though the manuscript is written in understandable English, the writing is short of publication standards in English.
Author Response
Ref: Submission ID healthcare-2748282: “Pain prevalence and satisfaction with pain management in inpatients: a cross-sectional study”
Reviewer #2
Q1) Peer review report for the manuscript entitled “Pain prevalence and satisfaction with pain management in in-patients: a cross-sectional study.” The study determined the pain prevalence in a tertiary university hospital in Spain, the risk factors of the pain, and patient satisfaction with the hospital’s pain management. The results showed that pain prevalence among adult in-patients was high, no risk factor examined was significantly associated with pain, and patients were satisfied with the hospital’s pain management despite the high pain intensity.
R1.- We thank the Reviewer for the comments about our manuscript. We appreciate the effort of reading it so carefully and for the insightful views on the matter.
Q2) I have suggestions and comments to improve the manuscript and hope the authors will consider them. 1. Line 14. I suggest the authors clarify the “satisfaction” with the hospital’s pain management.
R2.- Thank you for this suggestion. As suggested, this sentence has been modified: “The aim of this study was to evaluate pain prevalence and satisfaction with the hospital’s pain management in patients attended at a tertiary university hospital.”
Q3) 2. Line 25. “A problem yet to be solved” gives me the impression that the hospital has tried to solve the issue but has yet to succeed. However, the current study actually just explored the pain prevalence. Thus, I suggest rephrasing to accurately conclude from the study, which showed that the pain prevalence is high. (While hospitals worldwide strive to provide “pain-free” environments, 52.9% is highly prevalent.)
R3.- We thank the Reviewer for this suggestion. We have rephrased this sentence. Now it states: The prevalence of pain in hospitalized patients was high, despite the fact that patient satisfaction was also very high.
Q4) 3. Lines 53 – 55. I do not understand the sentence, so I suggest rephrasing it.
R4.- Thank you. As suggested, we have modified the sentence. Now it states as follows: As a secondary objective, we evaluated the percentage of patients with pain who were receiving analgesics. We also analyzed different risk factors that may have influenced pain during the hospital stay.
Q5) 4. The Introduction Section needs improvement. Pain management has been studied around the globe for the past decades. Why is it crucial for the authors to conduct the study?
R5.- We consider that adequate pain management is mandatory in hospitalized patients. Therefore, it is necessary to monitor how many patients suffer from pain. As requested by the Reviewer, we have added the following to the Introduction section: Pain in hospitalized patients is, and should be, one of the main concerns from a therapeutic point of view.
Q6) 5. Line 63. Perhaps the authors can briefly describe (1 or 2 sentences) STROBE guidelines, such as what STROBE stands for and its uses and aims.
R6.- Thank you. We have added a little explanation describing the STROBE initiative: The STROBE initiative establishes recommendations regarding the items that should be addressed in articles reporting cross-sectional studies.
Q7) 6. Lines 68 – 69. The sentence is a duplicate of what is in lines 63 – 64.
R7.- Thank you for this comment. We have deleted the duplicated sentence.
Q8) 7. Lines 77 – 80. I suggest rephrasing for clarity. (Is the horizontal line 10 cm in length?)
R8.- Thank you for this comment. We have rephrased the sentence. Now it states as follows: The Visual Analogue Scale (VAS) was employed to evaluate pain intensity, from 0 (no pain) to 10 (maximum pain) based on the patient indicating a point on a horizontal line of 10 cm in length representing his experience of pain.
Q9) 8. Lines 85 – 86. Perhaps the authors should provide the reason for NRS to be included. (qualitative & quantitative data?)
R9.- Thank you for this comment. We have added the appropriate reason for NRS to be included. Now it states: Satisfaction was measured not only as a dichotomous variable (yes/no), but also according to a Numerical Rating Scale (NRS) from 0 (no satisfaction) to 10 (maximum satisfaction). So, data can be evaluated not only from a qualitative, but also from a quantitative point of view.
Q10) 9. Line 102. I suggest rephrasing for clarity. (Have 544 patients been hospitalized for more than 24 hours or recently admitted within 24 hours?
R10.- We truly appreciate the suggestion. Now it states as follows: Of the 544 patients admitted for more than 24 h on the abovementioned day, the survey was done on 274.
Q11) 10. The tables may need to be rearranged for better visualization.
R11.- We appreciate the observations made by the reviewer. We hope that, after acceptance, tables will take the appropriate place (according to Editor’s decision).
Q12) 11. Line 136. “Thus” does not seem to be the right word.
R12.- Thank you for this suggestion. We have corrected the connector.
Q13) 12. Line 166. The in-text citation.
R13.- Thank you. We have modified it.
Q14) 13. Lines 171 – 173. The authors may need to follow up with the sentence. What is the purpose of this study? what is the population?
R14.- Thank you. The purpose of this study was already stated in the end of the introduction section: Our main objective was to evaluate the prevalence of pain in patients admitted to a tertiary university hospital. We also evaluated the satisfaction of patients regarding pain management during their hospital stay.
As requested, we added an explanation in the Material and methods section, so the potential reader can understand the characteristics of the population attended by the Hospital: The hospital in which the study was carried out is a tertiary university hospital located in a provincial capital. This serves as a reference for the following medical specialties: internal medicine, cardiology, pulmonology, endocrinology, nephrology, rheumatology, digestive, neurology, medical oncology, radiation oncology, geriatrics, hematology and psychiatry. It also has the following surgical specialties: vascular surgery, cardiovascular surgery, otorhinolaryngology, plastic surgery, general and digestive surgery, thoracic surgery, neurosurgery, ophthalmology, traumatology and orthopedic surgery, maxillofacial surgery, and urology. All named specialties are capable of training new specialists through an accredited system of specialized health training. As it is a hospital that is part of the Spanish public health system, the sociocultural and educational characteristics of the population served are heterogeneous.
Q15) 14. Line 174. What is the acceptable prevalence of pain, then?
R15.- Thank you for this reflection. In our opinion, “No patient under medical/healthcare supervision should experience pain. In addition, those patients should receive pain relief to minimize the intensity of the pain as much as possible.” We have added this information to the Discussion section.
Q16) 15. Line 175 – 177. This statement seems to belong in the introduction section. Why it is what it is to convince the importance of this study.
R16.- Thank you for this comment. As suggested, we have removed this statement from the Discussion section and added it to the Introduction.
Q17) 16. Lines 185 – 210. I do not see this paragraph as relevant. The patients who are best candidates for other methods are excluded from this study.
R17.- Thank you. We have erased this paragraph as requested.
Q18) 17. Line 211. I suggest replacing “complained” with “reported.”
R18.- Thank you. We have modified it as suggested.
Q19) 18. Lines 216 – 218. Perhaps this paragraph presented one of the study's limitations. while pain treatment is not one-size-fits-all, the method to measure pain intensity is.
R19.- Thank you for this comment. We have added this sentence in the Limitation section as suggested.
Q20) 19. Lines 225 – 226. I don't think the statement is valid unless the questions survey the pain improvement with treatment rather than the pain present.
R20.- We agree with the Reviewer. We have removed it as suggested.
Q21) 20. Line 243. Suggest replacing “determined” with “found.”
R21.- Thank you. We have modified it as suggested.
Q22) 21. Lines 256 – 261. Suggest rearranging the sentence. The sentence from lines 257 through 259 may be better following the sentence from lines 260 through 261—the second limitation.
R22.- Thank you for this comment. We have modified it as suggested.
Q23) 22. Lines 260 – 269. The justification for the second limitation is lengthy and unreasonable. Without validation, perhaps, the questionnaire should be piloted before being distributed.
R23.- Thank you for this comment. We acknowledge that piloting the questionnaire would have been ideal. However, as explained in the Limitations section, “the pragmatic perspective of the study made it necessary for the questionnaire to be based on usual clinical practice. Therefore, the questionnaire used was simple enough to be generalizable and understandable by all patients. Questions included in the survey have content validity by nature. Specifically, the questions about pain intensity and satisfaction with the treatment received are entirely subjective and there is little possibility that they can be answered in any other way.”
We have also added the following information regarding the methodology to the Limitations section: “Anesthesiologists performing the survey have extensive experience in pain management and accustomed to clinical interviewing, ensuring the best clinically available conditions”.
Q24) 23. Line 275. “Unsolved problem”. I refer to my comment on this phrase within the abstract.
R24.- Thank you for this suggestion. We have changed this phrase accordingly. Now it states as follows: “Pain in hospital patients is an important issue, even if patients are satisfied.”
Round 2
Reviewer 1 Report
Comments and Suggestions for Authors
Thank you for submitting this revised version of this paper. The revisions have significantly improved the manuscript, although there continue to be some methodological flaws in the study as it was conducted.
Line 106 "Therefore, data can be evaluated not only from a qualitative, but also from a quantitative point of view" but there is no mention of verbal descriptors. Qualitative data is about words and their meaning, and in this instance, patients were not asked to describe their pain, but to give a number - this is quantitative.
Line 203-205 states "No patient under medical/healthcare supervision should experience pain. In addition, those patients should receive pain relief to minimize the intensity of the pain as much as possible. " This does not appear to be supported by what is known about pain. Those with chronic pain in particular may not respond to any form of analgesia, and there are groups of pain phenotypes that do not respond according to the WHO analgesic ladder - eg neuropathic and nociplastic pains. Acknowledging that even with the best care, many people in hospital, including those in post-operative care, will continue to experience pain is crucial for establishing realistic expectations. I think this sentence should either be reworded to read 'All patients under medical/healthcare supervision should have their pain assessed and managed following evidence-based practice.'
The comments about patients being influenced by the presence of a medical practitioner in terms of their satisfaction rating is acceptable, although it may be more accurate to say that their reporting may be influenced rather than their actual satisfaction (demand characteristics).
Line 273 The authors state that "While pain treatment is not one size-fits-all, the method for measuring pain intensity is." but there are several different ways to measure pain intensity, including numeric rating scales, visual analogue scales, the 'faces' scale, 'pain thermometer' etc. This should be amended.
Comments on the Quality of English Language
There are some typographical errors that need careful proof-reading.
Author Response
Reviewer #1 (Round 2)
Q1) Thank you for submitting this revised version of this paper. The revisions have significantly improved the manuscript, although there continue to be some methodological flaws in the study as it was conducted.
R1.- We deeply appreciate the positive comments of the revised manuscript. We believe that we have followed and implemented all the recommendations and suggestions requested.
Q2) Line 106 "Therefore, data can be evaluated not only from a qualitative, but also from a quantitative point of view" but there is no mention of verbal descriptors. Qualitative data is about words and their meaning, and in this instance, patients were not asked to describe their pain, but to give a number - this is quantitative.
R2.- Thank you very much for this comment. We acknowledge that this point could have been explained better to make it clearer. We have thus modified it in the new revised version of the manuscript. Now it states as follows:
“Therefore, both data on pain and patient satisfaction with the hospital's pain management can be analyzed from a qualitative point of view: patient with or without pain, and patient satisfied or not. By using these scales, data can also be analyzed from a quantitative point of view: in terms of pain using the VAS scale from 0 to 10, and in terms of satisfaction using the NRS from 0 to 10.” (Lines: 106 – 110).
Q3) Line 203-205 states "No patient under medical/healthcare supervision should experience pain. In addition, those patients should receive pain relief to minimize the intensity of the pain as much as possible. " This does not appear to be supported by what is known about pain. Those with chronic pain in particular may not respond to any form of analgesia, and there are groups of pain phenotypes that do not respond according to the WHO analgesic ladder - eg neuropathic and nociplastic pains. Acknowledging that even with the best care, many people in hospital, including those in post-operative care, will continue to experience pain is crucial for establishing realistic expectations. I think this sentence should either be reworded to read 'All patients under medical/healthcare supervision should have their pain assessed and managed following evidence-based practice.'
R3.- Thank you very much for this recommendation. Following the suggestion, we have modified this sentence. Now it states:
“All patients under medical/healthcare supervision should have their pain assessed and managed following evidence-based practice. Thus, both the evaluation and treatment of pain must be carried out on an individualized basis with the aim of minimizing the patient's painful experience as much as possible.” (Lines: 206 – 210).
Q4) The comments about patients being influenced by the presence of a medical practitioner in terms of their satisfaction rating is acceptable, although it may be more accurate to say that their reporting may be influenced rather than their actual satisfaction (demand characteristics).
R4.- Thank you very much for this comment. Following the Reviewer’s suggestion, we have rephrased the sentence as follows: “The presence of a doctor during the survey, although not the treatment doctor, could have influenced the patients’ report about their level of satisfaction.” (Lines: 265 – 267).
Q5) Line 273 The authors state that "While pain treatment is not one size-fits-all, the method for measuring pain intensity is." but there are several different ways to measure pain intensity, including numeric rating scales, visual analogue scales, the 'faces' scale, 'pain thermometer' etc. This should be amended.
R5.- Thank you very much for this comment. We acknowledge that this sentence may be misunderstood by potential readers. Following your suggestion, we have modified the sentence. The following is now stated:
“While pain treatment is not one-size-fits-all, the need to use a method to measure pain intensity should be mandatory for everyone. In addition, clinicians should adapt the pain assessment, using the most appropriate tool for the patient's clinical situation.” (Lines 278 – 281).